# Iridium-Functionalized Cellulose Microcrystals as a Novel Luminescent Biomaterial for Biocomposites

**DOI:** 10.3390/biom12091165

**Published:** 2022-08-23

**Authors:** Mirko Maturi, Chiara Spanu, Andrea Baschieri, Mauro Comes Franchini, Erica Locatelli, Letizia Sambri

**Affiliations:** 1Department of Industrial Chemistry “Toso Montanari”, University of Bologna, Viale Risorgimento 4, 40136 Bologna, Italy; 2Institute for Organic Synthesis and Photoreactivity, National Research Council of Italy (ISOF-CNR), Via Gobetti 101, 40129 Bologna, Italy

**Keywords:** cellulose microcrystals, bronsted-acid ionic liquid, iridium complex, phosphorescence

## Abstract

Microcrystalline cellulose (MCC) is an emerging material with outstanding properties in many scientific and industrial fields, in particular as an additive in composite materials. Its surface modification allows for the fine-tuning of its properties and the exploitation of these materials in a plethora of applications. In this paper, we present the covalent linkage of a luminescent Ir-complex onto the surface of MCC, representing the first incorporation of an organometallic luminescent probe in this biomaterial. This goal has been achieved with an easy and sustainable procedure, which employs a Bronsted-acid ionic liquid as a catalyst for the esterification reaction of -OH cellulose surface groups. The obtained luminescent cellulose microcrystals display high and stable emissions with the incorporation of only a small amount of iridium (III). Incorporation of MCC-Ir in dry and wet matrices, such as films and gels, has been also demonstrated, showing the maintenance of the luminescent properties even in possible final manufacturers.

## 1. Introduction

Bio-based feedstocks, also derived from food and agriculture waste materials, represent extremely valuable tools as starting materials to produce more value-added derivatives. Amongst these, cellulose offers great opportunities because it can form densely packed crystalline domains thanks to intra- and intermolecular H-bonding. Such crystalline domains may form upon acid hydrolysis of micro- and nano-sized cellulose crystals generally referred to as microcrystalline cellulose (MCC) and cellulose nanocrystals (CNCs), respectively [1,2]. Due to the intrinsic nature of cellulose combined with the features of microstructured materials, MCC show distinctive chemical and physical properties that make them a good candidate for applications ranging from nanomedicine to materials science [3]. As previously reported, MCC has proven to be able to efficiently adsorb and release drugs for biomedical applications and tissue engineering, and its applicability also expands towards the packaging [4], cosmetics [5], and paper [6] industries, additive manufacturing [7], catalysis [8], and sensors [9]. Its properties include biocompatibility, high surface area, and low density, together with improved mechanical, thermal, and optical features. MCC represents therefore one of the most promising materials to replace plastic and metal functional parts in a great range of industrial and medical scenarios, without losing competitiveness in the market. In addition, the reactive hydroxy groups that stabilize the MCC surface offer the possibility for chemical modification, allowing for the opportunity to tune the physical-chemical properties of the material for specific applications [10]. Many functionalization methods have been developed and reported in the literature [11,12]. This has been achieved with several different synthetic chemical approaches, mainly including oxidation [13] or esterification with acyl chlorides and anhydrides [14]. Recently more sustainable strategies for cellulose -OH esterification have been achieved, such as the exploitation of ionic liquids that display intrinsic Bronsted acidity [15,16].

During recent decades, light-emitting materials generated widespread interest for various applications in different fields, from biochemical probes to the development of flexible, foldable, and luminescent devices, from soft materials to anti-counterfeiting technologies [17,18]. In particular, luminescent cyclometalated iridium(III) complexes display improved phosphorescence quantum yields and enhanced stability and versatility, which makes them suitable for applications ranging from solid-state devices to bioimaging and photocatalysis [19]. Such versatility arises from the possibility of finely tuning their photophysical properties by changing the nature and structure of the ligands, that can be prepared following various synthetic approaches [20]. The incorporation of iridium complexes into solid matrices and substrates can be challenging due to their limited water solubility and high cost, while their photoluminescent properties can suffer from high Ir(III) concentrations due to the occurrence of quenching phenomena [21]. The possibility to lock such complexes on the surface of a versatile biomatrix, while ensuring their dilution to minimize quenching effects, could represent a remarkable step toward new functional biobased platforms employing metal-based phosphors for a wide range of applications, ranging from bioimaging to luminescent soft materials like luminescent paper and anti-counterfeiting additives [22,23].

To the best of our knowledge, cellulosic materials have not been coupled to luminescent organometallic complexes, and the possibility of combining the outstanding properties of the two materials is a topic that is still completely missing from the scientific literature. Thus, this work aims to develop luminescent MCC through a simple reaction that takes advantage of the catalytic activity of a Bronsted-acid ionic liquid, namely, *N*-methyl pyrrolidinium hydrogensulphate, towards the esterification of the hydroxyl groups present on the surface of the cellulose microcrystals, with a bifunctional pyridyl triazole (pyta) ligand bearing a reactive carboxylic acid moiety. The pyta ligand is therefore employed for the complexation of Ir(III) using its cyclometalated dimer [Ir(ppy)_2_(*µ*-Cl)]_2_ (where Hppy = 2-phenylpyridine) as the precursor, leading to a surface-modified MCC with remarkable light-emission properties as well as enhanced photostability with very low heavy metal loadings. In addition, the luminescent micro-powder is formulated into wet and dry 3D biomatrices to demonstrate its possible use for a large range of different applications. 

## 2. Materials and Methods

All chemicals were purchased from Merck KGaA (Darmstadt, Germany) and used as received. All aqueous solutions were prepared with deionized water obtained using an ultrafiltration system (Milli-Q, Millipore, Burlington, MA, USA) with a measured resistivity above 18 MΩ/cm. 

### 2.1. Desulfonylation of MCC-OSO_3_^−^

Synthesis of sulfonated microcrystalline cellulose (MCC-OSO_3_^−^) was performed by acid hydrolysis of pure cellulose Whatman filter paper and desulphonated by adapting the approach firstly reported by Roman et al. in 2010 [24]. In a common procedure, 3.2 g of freeze-dried MCC-OSO_3_^−^ was placed in a 250 mL round-bottomed flask and dispersed in 100 mL of DMSO under continuous stirring. Once a homogeneous suspension was obtained, 190 μL of pyridine (2.3 mmol) was added and the mixture was kept under stirring at room temperature for 30 min. Then, the pyridine was removed through evaporation in a vacuum and the solution was emulsified by tip-probe sonication (40% amplitude, 3 min). Finally, 35 mL of methanol was added, and the mixture was refluxed for 2 h. The MCC-OH suspension was purified by three repeated centrifugations (7000 rpm, 5095× *g*, 20 min) and redispersion in water, followed by extensive dialysis (MWCO 3 kDa) against DI water. The purified MCC-OH was then freeze-dried and stored at room temperature. 

### 2.2. Synthesis of N-Methylpyrrolidinium Hydrogensulphate [MePyrrH][HSO4]

In a 500 mL round-bottomed flask, *N*-methylpyrrolidine (1.0 mol, 104 mL) was cooled under stirring to 0 °C with an ice bath, then H_2_SO_4_ (1.0 mol, 53.3 mL) was slowly added dropwise. After this addition, the mixture was kept under stirring at 60 °C for 30 min to ensure the complete conversion of the reactants into the ionic liquid, which was then cooled to room temperature and stored at +4 °C before use, without further purification.

### 2.3. Synthesis of 7-(4-(Pyridin-2-yl)-1H-1,2,3-triazol-1-yl)heptanoic Acid (pyta-COOH) and 2-(1-Nonyl-1H-1,2,3-triazol-4-yl)pyridine (pyta-CH_3_)

The COOH-bearing pyridyl triazole ligand pyta-COOH was prepared as previously reported by Sambri and co-workers in 2015, with small modifications [22]. Concerning the procedure described by Naddaka et al., in this work ethyl 7-(4-(pyridin-2-yl)-1*H*-1,2,3-triazol-1-yl)heptanoate (pyta-COOEt) underwent alkaline hydrolysis before its complexation to the Ir(III) dimer. In a typical procedure, 175 mg (0.58 mmol) of pyta-COOEt was dissolved in 35 mL of DCM, and a solution of 560 mg (10 mmol) of KOH in 15 mL of EtOH/H_2_O 1:1 was slowly added. The saponification reaction was performed by stirring the mixture overnight at room temperature. The next day, the organic solvents were removed by rotary evaporation, and the aqueous mixture was brought to neutral pH using HCl 1 M. Finally, the product was extracted three times from the aqueous phase using 20 mL of DCM for each extraction, which was then dried over sodium sulfate and evaporated to leave 7-(4-(pyridin-2-yl)-1*H*-1,2,3-triazol-1-yl)heptanoic acid as a white solid (Yield 64%). ^1^H NMR (600 MHz, Chloroform-*d*) δ 9.60 (s, 1H), 8.75 (d, 1H), 8.72–8.68 (m, 1H), 8.41 (m, 1H), 7.75 (t, 1H), 4.52 (t, 2H), 2.31 (t, 2H), 2.02 (p, 2H), 1.67 (p, 2H), 1.41 (m, 2H), 1.34 (m, 2H).

Analogously, 2-(1-nonyl-1*H*-1,2,3-triazol-4-yl)pyridine (pyta-CH_3_) was prepared using 1-bromononane as a starting material instead of ethyl 7-bromoheptanoate (Appendix A). ^1^H NMR (600 MHz, Chloroform-*d*) δ 8.56 (d, 1H), 8.17 (d, 1H), 8.13 (s, 1H), 7.77 (t, 1H), 7.21 (m, 1H), 4.40 (t, 2H), 1.94 (m, 2H), 1.32 (m, 4H), 1.25 (m, 8H), 0.85 (t, 3H).

### 2.4. Esterification of MCC-OH with Pyta-COOH

Typically, 1.5 g of MCC-OH was dispersed in 60 mL of water by tip-probe sonication (40% amplitude, 3 min) and then added to a 500 mL round-bottom flask containing pyta-COOH (300 mg, 1.09 mmol) in a mixture of 60 mL of MeOH and 60 mL of [MePyrrH][HSO_4_]. The excess methanol was eliminated by rotary evaporation, then the mixture was kept under stirring at 60 °C in a high vacuum for 6 h. Purification was performed by three repeated centrifugations (7000 rpm, 5095× *g*, 20 min) and redispersion in 50 mL of water, followed by extensive dialysis (MWCO 3 kDa) against DI water. The final suspension was then freeze-dried to obtain MCC-pyta as a dry white powder.

### 2.5. Synthesis of MCC-Ir and [Ir(ppy)_2_(pyta-COOH)]Cl

The MCC-pyta (250 mg) was dispersed in 7.5 mL of ethanol by tip-probe sonication (40% amplitude, 3 min), then 50 mg of [Ir(ppy)_2_(μ-Cl)]_2_ (46.6 mmol) in 22.5 mL of dichloromethane was added dropwise. The solution was stirred overnight at room temperature, after which the iridium conjugate was purified by repeated centrifugation (7000 rpm, 5095× *g*, 20 min) and redispersion in 10 mL DCM until colorless supernatant was obtained (three washes). A final washing step was performed with a 5% solution of NH_4_PF_6_ in 1:3 EtOH/DCM to remove all adsorbed free cationic iridium complexes from the cellulose microcrystals. MCC-Ir was collected, dried from DCM, and redispersed in 5 mL of water before freeze-drying. The free unbound [Ir(ppy)_2_(pyta-COOH]Cl complex was similarly prepared as previously described above [22].

### 2.6. Materials Characterization

Centrifugation was performed on a Sartorius G-16 centrifuge equipped with a YCSR-A0B fixed-angle rotor. For tip-probe sonication, a Sonics VibraCell (Newtown, CT, USA) ultrasound generator was employed. Fluorometric analysis was performed with an Edinburgh FLSP920 (Livingston, UK) spectrofluorometer equipped with a 450 W Xenon arc lamp. ^1^H-NMR spectra were obtained on a Varian Inova (600 MHz) NMR spectrometer (Palo Alto, CA, USA). In all recorded spectra, chemical shifts are reported in ppm of frequency relative to the residual solvent signals for CDCl_3_ (^1^H: 7.26 ppm). ATR-FTIR analysis was performed using a Cary 630 FTIR spectrometer (Agilent, Santa Clara, CA, USA). SEM images were acquired with a field emission gun scanning transmission electron microscope ZEISS LEO Gemini 1530 (FEG-STEM) (Jena, Germany). Powder XRD analysis was performed with a vertical goniometric diffractometer Philips PW 1050/81 (Eindhoven, The Netherlands) with Bragg–Brentano geometry equipped with a PW 1710 chain counting employing Cu-Kα radiation. ζ-potential measurements were performed using a Malvern (Malvern, UK) Zetasizer Nano-S with a 532 nm laser beam in DTS1070 clear disposable zeta cells at 25 °C. The iridium content of the preparation was evaluated by ICP-MS (Agilent 7850) after acid microwave digestion of a portion of each sample.

### 2.7. Photostability of MCC-Ir

To assess the photostability of the MCC-Ir conjugate, the freeze-dried powder was irradiated in a UV chamber (Sharebot CURE, Nibionno, Italy, λ = 375−470 nm, 120 W) for 30 min during which the sample was collected at different time steps and its luminescent behavior was analyzed by spectrofluorometry. At the same time, to compare the photostability of MCC-Ir with the free complex dispersed in an inert matrix, PMMA films were prepared by dissolving 763 μg (940 nmol) of [Ir(ppy)_2_(pyta-COOH)]Cl together with 1 g of PMMA (Mw = 350,000) in 50 mL of DCM. Then, 10 mL of the as-prepared solution was poured into a 5-cm glass Petri dish and the solvent was evaporated at room temperature to give transparent Ir-loaded thin PMMA films. A sample of 1 cm^2^ was cut and irradiated in the same conditions described for MCC-Ir and its luminescent behavior was analyzed over time by spectrofluorometry. Since the UV chamber is internally totally reflective and thus, since the sample is the only UV-absorbing material, all the 120 W fell on the sample and the irradiance in the UV chamber was approximated to the ratio between the source power (120 W) and the sample area (~1 cm^2^), therefore equal to 120 W/cm^2^.

### 2.8. Oxygen Sensing Capability of MCC-Ir

To evaluate the response of the material to the presence of molecular oxygen in terms of phosphorescence emission intensity, 1.5 mg of MCC-Ir was dispersed in 3 mL of water and placed in a gas-tight quartz cuvette equipped with a rubber septum. The emission behavior of the dispersion was analyzed by detecting the emitted light in the 450 to 650 nm range by excitation at 310 nm. Emission spectra were recorded before and after 10 and 20 min of N_2_ bubbling into the solution.

### 2.9. Formulation of MCC-Ir-Loaded Cellulose/Agarose (MCC/Agarose) Hydrogels

In a beaker, 50 mg of agarose and 50 mg of MCC-Ir were dispersed in 1 mL of water by heating under continuous stirring until a homogeneous dispersion was obtained. Then, the mixture was quickly transferred into a 24-well plate and cooled to room temperature to allow for the hardening of the agarose matrix, leading to a hydrogel containing 5 wt.% of MCC-Ir. Lower MCC-Ir concentrations (1 wt.% and 2 wt.%) were achieved by replacing part of the MCC-Ir with MCC-OH.

### 2.10. Solvent-Casting of MCC-Ir-Loaded Cellulose/Xanthan Gum (MCC/XG) Thin Films

In a round-bottomed flask, 25 mg of MCC-Ir and 275 mg of MCC-OH were dispersed in 10 mL of an aqueous solution containing 150 mg of glycerol and 50 mg of xanthan gum (XG). The solution was stirred at 70 °C until a homogeneous dispersion was obtained. This mixture was poured into a petri dish (3.5 × 3.5 × 4 cm) and then dried in an oven at 80 °C for 10 h, leading to a thin film containing 5 wt.% of MCC-Ir. Lower MCC-Ir concentrations (1 wt.% and 2 wt.%) were achieved by replacing part of the MCC-Ir with MCC-OH.

## 3. Results

The reactive surface hydroxy groups of MCC offer the possibility to introduce alternative functional moieties via an esterification reaction [10,14,25]. In the present work, the surface of MCC was decorated with the multifunctional ligand pyta-COOH, properly designed to be linked both to the MCC and to the luminescent Iridium-core. In particular, pyta-COOH contains a carboxylic acid able to form esters with the OH-groups and a 2-pyridyl triazole that, working as ancillary ligand, can allow for the formation of a luminescent pendant Ir(III) complex, namely Ir(ppy)_2_(pyta-COOR), [pyta = 4-(pyrid-2′-yl)-1,2,3-triazole, Hppy = 2-phenylpyridine)] stably bound to the biomaterial’s surface.

In detail, the ester pyta-COOEt, which we already employed to covalently bind iridium (III) complex to amino-terminating micelles via amidation chemistry [22], was transformed in the corresponding carboxylic acid pyta-COOH, then bound to the MCC surface via esterification to give MCC-pyta, followed by the formation of the luminescent Ir(III)-derivative MCC-Ir by reaction with the dimer precursor [Ir(ppy)_2_(μ-Cl)]_2_ (Figure 1).

### 3.1. Synthesis and Characterization of MCC-Ir

Sulfonated cellulose microcrystals (MCC-OSO_3_^−^) were prepared by acid hydrolysis of cellulose filter paper mediated by sulfuric acid, as previously described by Jiang et al., but with small modifications [24]. Such an approach to hydrolysis leads to the formation of sulfate half-esters on the cellulose backbone, especially on part of the surface hydroxyl groups that are more exposed to the acid aqueous environment during the production of the microcrystals. This was also confirmed by the ζ-potential of the obtained micropowder, which reached values as low as −25.5 mV, compatible with net negative charges on the MCC surface. In order to increase the reactivity of the MCC surface towards esterification with carboxylic acids, MCC-OSO_3_^−^ was desulphonated using a solvolytic approach involving first a conversion of MCC-OSO_3_^−^ to its pyridinium salts in DMSO, followed by refluxing it in methanol for 2 h. This approach allowed for the regeneration of the reactive surface OH groups of cellulose microcrystals, as confirmed by the measured reduction in the ζ-potential from −25.5 mV of MCC-OSO_3_^−^ to −10.5 mV of MCC-OH due to the decrease of net negative charges. Then, the OH-rich MCC powder was subjected to esterification with pyta-COOH using the ionic liquid *N*-methyl pyrrolidinium hydrogensulfate ([MePyrrH][HSO_4_]) which acts both as the solvent and as Bronsted acid catalysts, as has been previously reported [16]. After thorough purification, the effectiveness of the surface functionalization of MCC with pyta-COOH was assessed by ATR-FTIR spectroscopy, which revealed the appearance of a peak at 1644 cm^−1^, which can be attributed to the stretching of the carbonyl in the ester linkage (Figure 2a). The low intensity of the IR absorption band is thought to be related to the fact that the esterification reaction is limited to the surface monomers of MCC which are exposed to the ionic liquid in the reaction mixture, while the crystalline core of MCC remains intact. In addition, a ζ-potential analysis revealed an average surface potential for MCC-pyta equal to −8.44 mV. The reduction in the negative surface potential is thought to be related both to the reduced content of free hydroxyl groups on the MCC surface compared to MCC-OH and to the presence of pyridine residues that are partially protonated in neutral aqueous environments. Then, a reaction was initiated betwee the pyridyl-triazole-bearing MCC-pyta and the cyclometalated iridium dimer [Ir(ppy)_2_(µ-Cl)]_2_, which is well known to interact with N^N chelating ligands, forming luminescent cationic complexes. Some of the unbound ligand, protonated by the Bronsted acidity of [MePyrrH][HSO_4_], is expected to be retained on the MCC surface after the esterification due to its negative surface potential, leading to the presence of free unbound pyridyl triazole ligand during the process of the conjugation to the iridium dimer. This would lead to the co-formation of free luminescent [Ir(ppy)_2_(pyta-COOH)]^+^ complexes in the presence of the Ir(III) dimer, which could remain adsorbed on the MCC surface by way of weak and unstable electrostatic interactions. The obtained MCC-Ir powder was therefore washed several times with a 5% solution of NH_4_PF_6_ in EtOH/DCM 1:3, which would ensure the complete replacement of mixed anionic counterions with hexafluorophosphate ions, leading to the removal of electrostatically adsorbed [Ir(ppy)_2_(pyta-COOH)]^+^ species. The ζ-potential of the obtained dispersion of MCC-Ir in water was then measured, and it was observed to increase from −8.44 mV for MCC-pyta to −31.8 mV of MCC-Ir, due to the presence of hexafluorophosphate counterions. Subsequently, the obtained MCC-Ir powder underwent a fluorometric analysis to assess its luminescent properties. Emission spectra of MCC-Ir and [Ir(ppy)_2_(pyta-COOH)]Cl and excitation spectra of MCC-Ir are reported in Figure 2b. The excitation spectrum confirms the possibility of luminescence in the range from 280 and 410 nm, and the emission spectra show the characteristic emission profile of phenyl pyridine cyclometalated Ir-complexes containing a pyridine 1,2,3-triazole as ancillary ligand with maximum peaks at 480 and 508 nm, by excitation at 310 nm, as previously reported [22]. The full emission map for MCC-Ir is reported in the Appendix A. The emission behavior of MCC-Ir at the solid state was compared to that of [Ir(ppy)_2_(pyta-COOH)]Cl, measured in the same experimental conditions. The emission profile recorded for the pure free complex was indeed similar to that of MCC-Ir, but due to intermolecular collisions and quenching phenomena, the emission intensity of the free complex is more than 100 times lower. This piece of evidence suggests that the conjugation of the Ir(III) complex on the MCC surface can reduce quenching phenomena, and strong emission intensities can be achieved with very low iridium loadings. In fact, the total iridium content of MCC-Ir measured by ICP-MS was equal to 180 ± 53 μg of iridium per gram of MCC-Ir, corresponding to 940 ± 270 nmol of iridium per gram of MCC-Ir. On the other hand, powder XRD analysis on MCC-Ir shows no significant differences in the diffraction pattern compared to pristine MCC or cellulose powder (Figure 2c), suggesting that the chemical modification of MCC was limited to the surface of the microcrystals, without affecting their crystallinity. Moreover, the morphology of the microcrystals was conserved after the surface functionalization, as revealed by the SEM images which show crystals of around 10 µm in length assembled in macrostructures (Figure 2d and Appendix A).

To prove the effective esterification of MCC with pyta-COOH, and therefore the stable covalent conjugation of the iridium complex to MCC surface, the same conjugation approach was employed using a similar ligand but bearing no carboxylic residues, named pyta-CH_3_ (Figure 3). This modified pyridyl triazole ligand was placed in the ionic liquid together with MCC-OH and exposed to the same experimental conditions reported for pyta-COOH, including the following step of iridium complexation. In this case, even though luminescence under UV light could still be observed after washing with DCM, a very weak luminescence was detectable after washing the MCC powder with NH_4_PF_6_ solution. Such observations suggested that MCC had been able to adsorb cationic species on their surface by way of weak electrostatic interactions that can be easily disrupted by washing MCC with an electrolyte solution. This was also confirmed by an ICP-MS analysis of an MCC sample treated with the non-reactive pyta-CH_3_ ligand, which revealed a total iridium concentration as low as 18 ± 8 μg of iridium per gram of MCC, corresponding to 94 ± 42 nmol of iridium per gram of MCC-Ir. Moreover, this experiment confirms the effectiveness of the esterification of MCC with pyta-COOH and its consecutive covalent functionalization with the Ir(III) cyclometalated complex.

### 3.2. Photostability of MCC-Ir

Cyclometalated iridium complexes are known for their intrinsic photostability compared to organic fluorophores, which allows them to efficiently maintain their photoluminescence after prolonged irradiation with UV light [26]. In order to evaluate the photostability of MCC-Ir, a portion of the freeze-dried micropowder was analyzed over time by spectrofluorometry during irradiation in a UV chamber working between 375 and 470 nm. At the same time, PMMA films loaded with [Ir(ppy)_2_(pyta-COOH)]Cl were prepared using a simple solvent casting approach: PMMA and the free iridium complex were co-dissolved in dichloromethane, then the mixture was poured in a Petri dish and dried at room temperature to achieve transparent luminescent films. The amount of loaded iridium was equal to 940 nmol of iridium per gram of PMMA, which is the same iridium concentration determined for MCC-Ir by ICP-MS. The film was subjected to the same test conditions to study the photostability of MCC-Ir in order to compare the decay profiles of their emission intensities during exposure to UV light.

Emission spectra were recorded by excitation at 310 nm, and the intensity of the highest emission peak (508 nm) was plotted for each tested sample against the corresponding radiant exposure at each time step, obtained by multiplying the exposition time by the radiant power emitted by the UV light source (Figure 4). In this way, the emission intensity is presented as a function of the total energy administered to the system per unit area, regardless of the radiant power of the employed light source.

By comparing the emission intensity decays for MCC-Ir and [Ir(ppy)_2_(pyta-COOH)]Cl, it is clearly observable that the decay profiles are very similar, suggesting that the covalent bond between the pyta ligand and the solid cellulose matrix does not affect the photostability of the system.

The emission intensities (I), measured as a function of the radiant exposure ($H_{e}$) during the photostability experiments, were fit using a double exponential equation (Equation (1)), which has been observed to efficiently model the photobleaching of iridium-based phosphors such as Ir(ppy)_3_ [27].
(1)$$I=I_{0}\cdot \left[\alpha \cdot e^{-k_{1}H_{e}}+\left(1-\alpha \right)\cdot e^{-k_{2}H_{e}}\right],$$

In Equation (1),$\;\;I_{0}$ represents the emission intensity at t = 0, and it has been constrained to a value of 1 for normalization. Moreover, a plateau of I = 0 at infinite radiant exposure was also set as a constraint for the fit since the emission from the complex is expected to be completely neutralized after an infinite amount of radiant energy is administered to the system. The results of the double exponential fittings are reported in Appendix A. By extrapolation of the fitting curve for MCC-Ir, a 90% reduction in the emission intensity is expected with a radiant exposure of around 30 KJ/cm^2^, which corresponds to around 500 to 1000 times the worldwide average UV daylight dose [28], or to the exposure for 300 s to the most commonly employed UV lasers for confocal microscopy (working around 100 W/cm^2^).

### 3.3. Oxygen Sensing Capability of MCC-Ir

The phosphorescence emission intensity of cyclometalted Ir-complexes is known to depend on the amount of oxygen in the sample due to the quenching phenomena. Therefore, materials containing such complexes can be applied as oxygen sensors.

As a proof of concept, we recorded emission spectra for MCC-Ir in an aqueous dispersion before and after the removal of oxygen from the solution via nitrogen bubbling after various lengths of time. As expected, and as reported in Figure 5, an increase in emissions was observed after 20 min, confirming the viability of employing the described material as an oxygen sensor since the emission intensity varies significantly with the change in the oxygen content of its surroundings.

### 3.4. Formulation of MCC-Ir in Dry and Wet 3D Matrices

The prepared MCC-Ir was therefore employed for the preparation of dry and wet 3D biomaterials in order to prove its applicability as a luminescent bio-additive for a wide range of applications. For the dry 3D matrix, we prepared thin films of MCC-OH and xanthan gum (XG) by way of solvent casting, adding glycerol as a plasticizing agent. Xanthan gum has been widely explored in the literature as a biocompatible and sustainable biomaterial for the production of thin films, whereas MCC has been blended with several polysaccharides to act as a reinforcing agent [29,30,31]. Furthermore, glycerol has been extensively investigated as a biocompatible plasticizer for biomaterial-based thin films and hydrogels [32,33]. The composition of the prepared film was 60% MCC, 10% XG, and 30% glycerol. Luminescent MCC/XG thin films were obtained by partially replacing MCC-OH with MCC-Ir up to 5% of the total dry weight. Similarly, wet 3D matrices were prepared using agarose, which is well-known to form solid hydrogels when cooled from its hot solutions [34], and MCC, which has been shown to promptly improve the mechanical properties of agarose hydrogels [35]. The composition of the gels was 5% agarose, 5% MCC, and 90% water. Phosphorescent hydrogels were prepared by partially replacing MCC-OH with MCC-Ir up to 5% of the total weight of the hydrogel. The prepared formulations were visibly luminescent under exposure to UV light, even with loadings of MCC-Ir as low as 1% of the total sample weight (Figure 6a). 

In addition, spectrofluorometric analyses performed on both films and gels showed no appreciable variation in the emission maximum of the phosphor, suggesting that the iridium emissive core was not being affected by the interaction with the biopolymers composing the samples, nor by the thermal treatments required for the manufacture of the films and hydrogels (Figure 6b).

## 4. Conclusions

In conclusion, we succeeded in developing a highly effective and stable luminescent probe, which combines iridium(III)-based organometallics with a completely bio-based matrix such as microcrystalline cellulose: the two moieties were linked together with a covalent bond, thus preventing the leaching of metal from the main structure. The obtained system maintains the exquisite properties of the luminescent agent as well as the integrity and thus the applicability of the MCC: this was demonstrated by the ability of the final compound to be used in the formation of gel and/or films which still display the same properties of the free iridium(III) luminescent complex. This will allow for the application of this compound in many different fields, such as biomedicine and bioimaging but also composite materials development.

In addition to the properties and uses previously described, this new material could be developed for its possible use as a photocatalyst for carrying out new organic reactions activated by light. The use of transition-metal complexes as visible-light homogeneous photoredox catalysts has rapidly grown and the demand for more efficient catalysts is continually increasing [36,37,38,39,40]. The process of recovery and recycling of organic catalysts through heterogenization has already been well developed but for organometallic compounds, a lot of work is still to be done [41]. The incorporation of catalysts in insoluble solid supports can provide highly active catalysts even in homogeneous catalysis, in combination with high recyclability. Heterogeneous catalysts can be recovered from the reaction mixture more straightforwardly, using simple techniques, such as filtration and centrifugation. Ongoing studies aim to use MCC-Ir for this purpose.

## Figures and Tables

**Figure 1 biomolecules-12-01165-f001:**
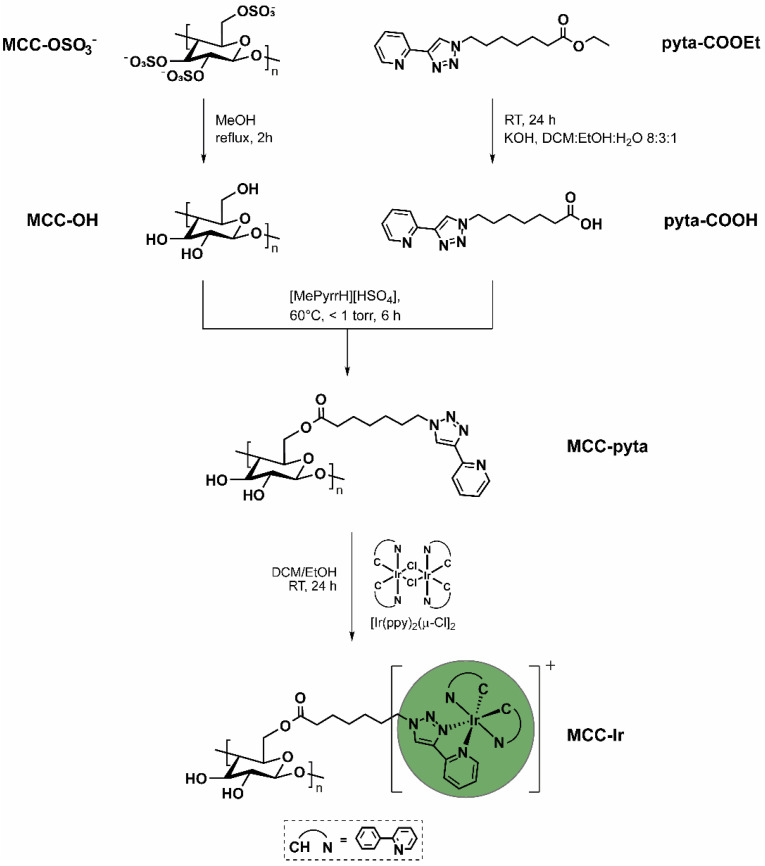
Synthetic approach for the conjugation of luminescent iridium (III) complex to the MCC surface starting from sulfated MCC and the previously reported pyridyl triazole ligand pyta-COOEt.

**Figure 2 biomolecules-12-01165-f002:**
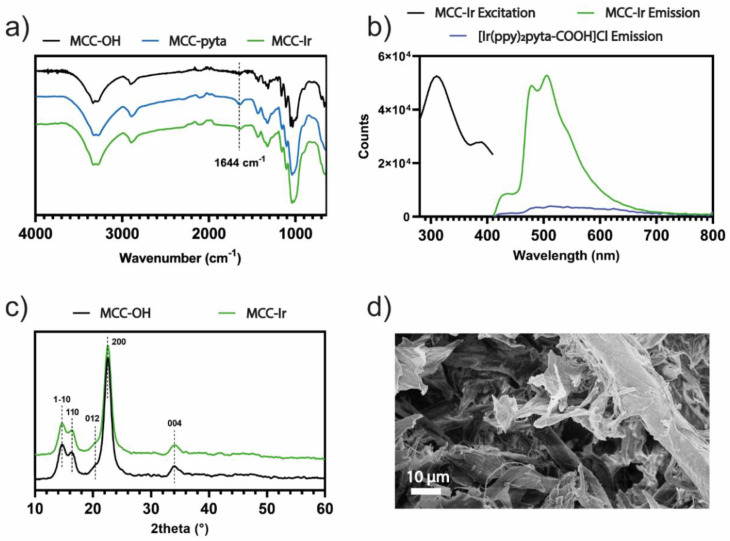
Spectroscopic and microscopic characterization of synthetic materials. (**a**) ATR-FTIR spectra of MCC-OH, MCC-pyta, and MCC-Ir. (**b**) Solid-state phosphorescence studies. For both MCC-Ir and [Ir(ppy)_2_(pyta-COOH)]Cl emission spectra, the excitation wavelength was 310 nm and the slits were set at 2 nm for both excitation and emission beams. For the excitation spectrum of MCC-Ir, the emission intensity at 508 nm was recorded by varying the excitation wavelength from 280 to 410 nm and the slits were set at 2 nm for both excitation and emission beams. (**c**) Powder XRD spectra of MCC-OH and MCC-Ir. The Miller indices attributed to the MCC’s X-ray reflections are reported above each peak. (**d**) Scanning electron microscopy (SEM) image of MCC-Ir.

**Figure 3 biomolecules-12-01165-f003:**
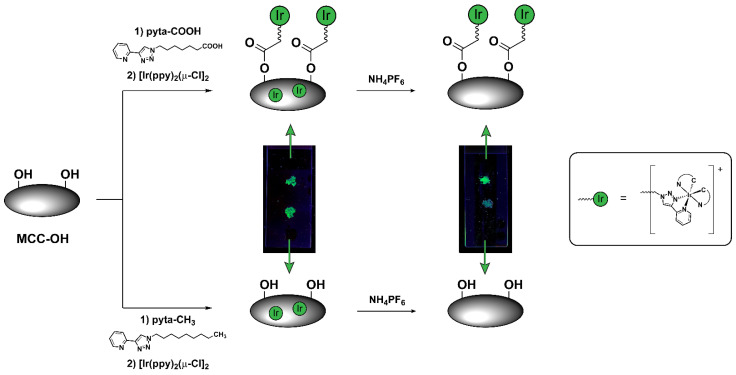
Electrostatic adsorption of unbound pyridyl triazole ligand and cationic cyclometalated complexes led to luminescent MCC micropowder even when treated with pyta-CH_3_, a ligand for Ir(III) that is not reactive towards MCC, as demonstrated by the digital camera pictures of MCC powders recorded under exposure to 365-nm light. However, such weak interactions are promptly disrupted by treating the powder with an electrolyte solution (5% NH_4_PF_6_ in EtOH/DCM), leading to a complete loss of luminescence for the sample treated with pyta-CH_3_. On the other hand, MCC covalently bound to the pyridyl triazole ligand via esterification maintain their luminescent properties after the treatment with NH_4_PF_6_ (right picture).

**Figure 4 biomolecules-12-01165-f004:**
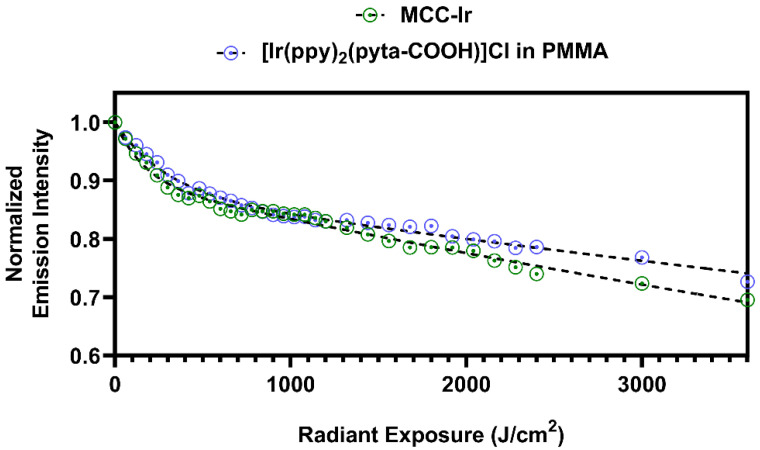
Photostability under UV irradiation of MCC-Ir compared to [Ir(ppy)_2_(pyta-COOH)]Cl. Emission spectra were recorded at different time points during the irradiation. The excitation wavelength was 310 nm, and the slits were set at 2 nm for both excitation and emission beams. The emission intensities at the highest peak (λ = 508 nm) are plotted against the corresponding radiant exposure and fit with double exponential decay functions (dashed line) according to Equation (1).

**Figure 5 biomolecules-12-01165-f005:**
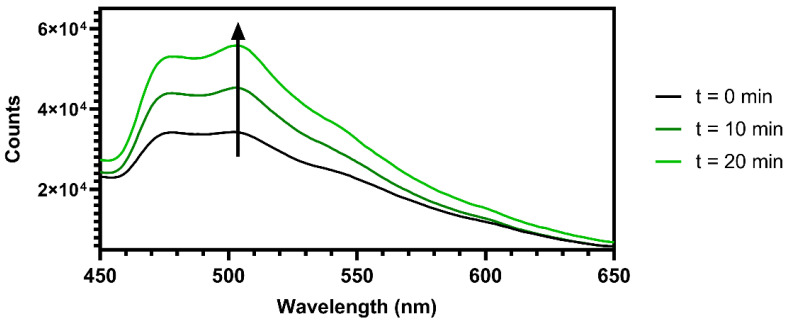
Oxygen sensing capabilities of MCC-Ir. The emission spectra of a suspension of MCC-Ir in water were recorded by excitation at 310 nm (slits 5 nm) after bubbling N_2_ gas into the solution for different amounts of time, revealing emission enhancement over time due to the removal of oxygen triplet species. The arrow indicates decreasing O_2_ content in solution.

**Figure 6 biomolecules-12-01165-f006:**
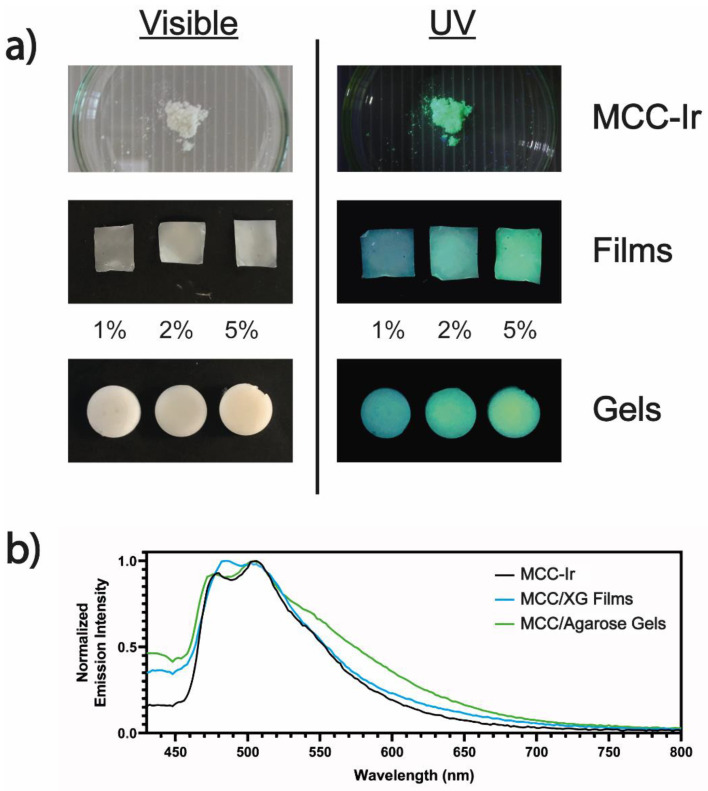
Manufacturing of films and gel containing Ir-loaded MCC. (**a**) Optical camera images of MCC-Ir, MCC/XG film, and MCC/Agarose hydrogels exposed to visible and UV light (λ = 365 nm). (**b**) Normalized emission spectra of MCC-Ir, MCC/XG thin films, and MCC/Agar hydrogels. For both acquisitions, the excitation wavelength was 310 nm and the slits were set at 2 nm for both excitation and emission beams.

## Data Availability

Not applicable.

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
