# Peer review of "Iridium-Functionalized Cellulose Microcrystals as a Novel Luminescent Biomaterial for Biocomposites"

_biomolecules, 2022, doi:10.3390/biom12091165_

Round 1

Reviewer 1 Report

Comments and recommendations on Biomolecules “Iridium-functionalized cellulose nanocrystals as a novel luminescent biomaterial for bio-nanocomposites”.

The authors describe the preparation and covalent conjugation of an Ir-complex to nanosized cellulose. The synthesis is described with sufficient detail but for a publication in a journal that is closely associated to molecules with biological use and/or function and an impact factor > 6, there is too little application perspective shown to justify publication in the current state. I therefore suggest publication after major revisions that should comprise at least two of the points that are suggested in my last comment. Other ideas by the authors are welcome, as well.

Further comments and recommendations, some of which are of major (*) importance:

- the introduction is a bit lengthy and could be shortened. Moreover, there are six self-citations of the last author in the article. Although some citations are necessary to refer to previous work, especially synthesis protocols, it is strongly advised to reduce self-citations to the lowest possible level.

- (*) the supporting information was missing and must be added for peer-review.

- Exptl. Section: please provide a list of devices and respective models used, e.g. sonication device, ICP-MS,

- section 2.1.: please give always centrifugal force in g rather than in rpm. Please give number of centrifugation steps. How can one estimate to have achieved sufficient purity to finish the centrifugations?

- section 2.3.: please number of extractions with DCM and volume of extraction per extraction step.

 - section 2.4.: I assume tip-probe sonification was used? Furthermore, please give centrifugal force in g rather than in rpm. Please give number of centrifugation steps. How can one estimate to have achieved sufficient purity to finish the centrifugations?

- section 2.5.: I assume tip-probe sonification was used?

- the two green spectra in figure 2A are hard to be discriminated. One spectrum in red color would be a better choice.

- (*) section 3.1.: the authors first claim that the addition of ammonium hexafluorophosphate would remove a noncovalently bound Ir-complex from the CNC surface but then find a percentage of 10 % to remain by ICP-MS analysis by the CNCs that were treated with the Ir-pyta-COOH complex. This is maybe documented by figure 3, where a blue spot is shown for this kind of CNCs. Aside from the fact that the caption of figure 3 does not indicate what is shown in the two black images with the dots (is it luminescence of the respective CNCs? what was the excitation source? is it a digital image of the CNCs taken under which conditions? on which surface with which parameters?) the authors cannot state in lines 310 and 311 that there is no luminescence detectable after washing the CNCs with ammonium hexafluorophosphate if there is still a blue spot visible in the bottom of the right black image of figure 3 (given that luminescence is shown). This statement that there is no luminescence detectable must be corrected and the origin of the blue spot must be explained and also why it is blue and not green. Moreover, I strongly recommend the authors to also the give emission spectra of the Ir-pyta-COOH complex. Maybe that would help to clarify this issue.

The authors comment that “the morphology of the nanocrystals was conserved after the surface functionalization, as revealed by the SEM images” but they do not describe the morphology, at all. This must be improved.

- section 3.2.: Please indicate the times that correspond to the various spectra shown in figure 4 left. It will give experimentalists a better orientation for bioanalytical applications.

- (*) section 3.3.: the embedding of the complex in films is nice, but gives no perspective on what could be the use of the luminescent CNCs in a biochemical/biomolecular/bioanalytical environment. This perspective must be added before the paper can be accepted because in the current state, the scientific content of the paper is too low to justify a publication in Biomolecules. Potential additional issues to be addressed to open a biomolecular perspective could be:

a) determine absorption and luminescence spectra of the films (of thinner films or some with lower CNC-content, provided scattering is too high) to show the suitability of the films to be analyzed by optical spectroscopic methods (e.g. photometry, luminescence spectroscopy, resonance light scattering or other method).

b) compare luminescence intensities (or even better quantum yields) of the complexes in the above films with the luminescence intensity of the complexes embedded in an oxygen-impermeable polymer. Alternatively, compare the quantum yields in an aquous buffer or in a buffer that was saturated with nitrogen. This could serve to open a perspective on oxygen sensing by luminescence quenching.

c) titrate the CNCs in aqueous solution with a standard protein like HSA or BSA to check whether the luminescent CNCs could be suitable for a general protein determination assay.

d) check if there is a luminescence response of the CNCs in solution (or of the films) upon addition of various amounts of DNA (single or double-stranded).

e) and finally, but very important: determine the pH-dependent stability of the covalent conjugation of the Ir-complex with cellulose. Maybe, pH-dependent luminescence of the Ir-complex is found as an additional merit of the complex.

Reviewer 2 Report

The manuscript is related to the synthesis and characterisation of nanocellulose with incorporated iridium (III) organometal complex. Full synthesis of the nanocellulose was given and its composition investigated by ATR-FTIR and z-potential analyses. The light emission properties of the obtained system by excitation with 310 nm light was given and emission stability on the UV irradiation was analysed. The study is important in the frame of biomaterials to increase the portion of recycled and biodegradable materials in production.

The paper is well written and could be accepted after minor correction.

11)      Line 106 "vacuo" should be changed to “vacuum”

22)      Line 158 Who is “some of us”

33)      Line 160 "Edimburg FLSP920" " should be changed to Edinburgh FLSP920

44)      Line 177-178 the 177 “irradiance in the UV chamber has been approximated to the ratio between the source power (120 W) and the sample area (~ 1 cm2), therefore equal to 120 W/cm2.” The statement is not clear. If all light is not irradiating on the sample, then the intensity that irradiate the sample will be lower. How it is in this case? All 120W is falling on the sample?

55)      English of all manuscript should be checked.

Author Response

The manuscript is related to the synthesis and characterisation of nanocellulose with incorporated iridium (III) organometal complex. Full synthesis of the nanocellulose was given and its composition investigated by ATR-FTIR and z-potential analyses. The light emission properties of the obtained system by excitation with 310 nm light was given and emission stability on the UV irradiation was analysed. The study is important in the frame of biomaterials to increase the portion of recycled and biodegradable materials in production.

The paper is well written and could be accepted after minor correction.

11)      Line 106 "vacuo" should be changed to “vacuum”

ANSWER:

Ok, done

22)      Line 158 Who is “some of us”

ANSWER:

We replaced “some of us” with “Sambri and co-workers”.

33)      Line 160 "Edimburg FLSP920" " should be changed to Edinburgh FLSP920

ANSWER:

Ok, done

44)      Line 177-178 the 177 “irradiance in the UV chamber has been approximated to the ratio between the source power (120 W) and the sample area (~ 1 cm2), therefore equal to 120 W/cm2.” The statement is not clear. If all light is not irradiating on the sample, then the intensity that irradiate the sample will be lower. How it is in this case? All 120W is falling on the sample?

ANSWER:

We understand that this point might have been not very clear, therefore we reformulated the sentence in paragraph 2.7 as follows, for better clarification:

Since the UV chamber is internally totally reflective and thus the sample is the only UV-absorbing material, all the 120 W are falling on the sample and the irradiance in the UV chamber has been approximated to the ratio between the source power (120 W) and the sample area (~ 1 cm2), therefore equal to 120 W/cm2.”

55)      English of all manuscript should be checked.

ANSWER:

Ok, done

Reviewer 3 Report

The manuscript from Maturi et al. presents the development of photoluminescent cellulose materials incorporating iridium (III) chelates as a principal luminophore. The authors discuss the synthesis and the basic characterization of the different components of the material and of the material itself.  IR spectra and electron microscopy images provide some evidence about the structural features and the morphology of the prepared iridium-modified cellulose. Photoluminescent films and hydrogels comprising this material serve as a proof-of-concept about its potential utility.  

The manuscript will be suited for publishing in Biomolecules after the authors address some of the issues with its current form.

1) The authors should remove the statements regarding the cost efficiency of the material they develop. While cellulose is, indeed, readily accessible, iridium is less abundant than gold and then platinum in the earth’s crust. Regardless of the fluctuations in its market price, the proof for the cost efficiency of technologies based on iridium is challenging to impossible.

2) The authors call the material cellulose nanocrystals (CNCs) or nanocellulose.

The EM image on Figure 2d, however, does not show any nanoparticle morphology.  CNCs-Ir looks more like an amorphous composition of microfibers.

Provide evidence for nanometer scale structures, such as images of the nanocrystals, general shapes, and size distribution (before modification and after each modification step).

3) One thing I cannot see in the main text is characterization of the iridium loading.

Once the authors characterize the nanocrystals (shapes and size distribution) they should determine and report the loading of Ir (III), e.g., how many moles of Ir per a gram of material and per a crystal.

It is crucial for understanding the utility of this material and for reproducibility of its preparation.

4) It is clear that the material is a UV absorber. Since the novelty of this material is its optical properties, the authors should place in the main text UV-visible absorption spectra of the CNC-Ir materials. Indeed, because of its light-scattering propensity (especially in the UV spectral region), acquiring of such spectra may require a spectrometer equipped with an integrating sphere, which is not always available. If it is the case, the authors can show the excitation spectra of the luminescent materials, which will give an idea where the light absorption leading to the formation of the emissive excited states is.    

5) As important as the photostability studies, they are not truly useful the way they are conducted and presented. Despite the analysis with eq. 1, the only conclusion from the results is that material (slowly) degrades under radiation comparable to that used in optical imaging techniques.

The authors need to complete proper actinometry studies and report the quantum yield(s) of degradation. Basically, carry out the studies along with a standard with known quantum yield of degradation and calculate from these results what is the quantum yield of CSCs-Ir degradation. The selection of the standard is important (it has to absorb in the same region and has comparable quantum yield of degradation).  

Round 2

Reviewer 1 Report

Comments and recommendations on Biomolecules “Iridium-functionalized cellulose nanocrystals as a novel luminescent biomaterial for bio-nanocomposites”, revised version.

I thank the authors for their efforts to considerably improving the manuscript. One comment was probably overlooked which is why it is added for another time. I additionally explain the reason for my last comment in my previous review for another time to convince the authors to make their paper more interesting to potential readers. I suggest publication after minor revision.

- section 2.3.: I cannot see where the requested information was added about the final extraction step. Please give number of extractions with DCM and volume of DCM per extraction step.

- section 3.3.: I acknowledge the addition of adding the emission spectra of the films by the authors. However, this only gives a weak glimpse on what could be a potential use of the luminescent CNCs in a biochemical/biomolecular/bioanalytical environment. The authors do not add any other of my suggestions because they suspect a lack of novelty as these applications are already published. It is true that these applications are already published but they should not be viewed under the aspect of bare novelty. My recommendation was directed to give potential readers nuclei for developing their own new application ideas by showing interesting additional properties of the complex like pH-dependent emission, oxygen-dependent emission, ability to noncovalently bind to certain biomolecules. It would be a lost chance to raise interest to the Ir-probe and luminescent CNCs by others. I therefore once again strongly recommend to choose one of the options I have outlined in my 1st report, particularly because they do not involve too much effort.

Reviewer 3 Report

The authors addressed most of the comments except a couple of relatively minor ones.

- The misuse of the term "crystalline  nanocellulose (CNC)".  While it is broadly used in the literature, it is not correct to use it when no nanometer-scale structural features are demonstrated. In addition, a broad use of a term does not make it right. Also, in the two publications that the authors cite as examples of using CNC for microstructures, actually there are images showing nanometer-scales features (fibers with thickness smaller than 100 nm or small 30-nm structures). The scale bar of the images that the authors of this manuscript show is 10 micrometers, while in the previously published reports some of the scale bars are 200 nm or so. 

I do not doubt the crystallinity of the structures. Nevertheless, it is not a proof for nanometer-scale dimensions.  While there may be alternative means to demonstrate nano-scale features of these cellulose structures, the authors are not presenting them.  

Removing nano and perhaps replacing it with micro will be the right thing to do for this work, to illustrate the nature of the the described materials.   Also, not calling it "nano" does not take away from the scientific merit of the work.  But calling it "nano" without evidence it is "nano" poses a question about the integrity of the presentation. 

- Figure S3 from the supporting information should be in the main text with a proper discussion.   Since describing the optical properties of these materials is a principal component of the manuscript, it is important to place evidence showing where the luminophore absorbs is crucial.

Author Response

Pleas see the attachment
